# Computationally reproducing results from meta-analyses in ecology and evolutionary biology using shared code and data

Steven Kambouris[1,2]*, David P. Wilkinson[1,2], Eden T. Smith[1,3], Fiona Fidler[1,2,3]

**1** MetaMelb Research Initiative, The University of Melbourne, Melbourne, Victoria, Australia, **2** School of Agriculture, Food and Ecosystem Sciences, The University of Melbourne, Melbourne, Victoria, Australia, **3** School of Historical and Philosophical Studies, The University of Melbourne, Melbourne, Victoria, Australia

* steven.kambouris@unimelb.edu.au

**Data Availability Statement:** The data and code files to reproduce all results reported in this article are available on Zenodo at https://doi.org/10.5281/zenodo.8114702. The reproducibility reports

## Abstract

Many journals in ecology and evolutionary biology encourage or require authors to make their data and code available alongside articles. In this study we investigated how often this data and code could be used together, when both were available, to computationally reproduce results published in articles. We surveyed the data and code sharing practices of 177 meta-analyses published in ecology and evolutionary biology journals published between 2015–17: 60% of articles shared data only, 1% shared code only, and 15% shared both data and code. In each of the articles which had shared both ($n = 26$), we selected a target result and attempted to reproduce it. Using the shared data and code files, we successfully reproduced the targeted results in 27–73% of the 26 articles, depending on the stringency of the criteria applied for a successful reproduction. The results from this sample of meta-analyses in the 2015–17 literature can provide a benchmark for future meta-research studies gauging the computational reproducibility of published research in ecology and evolutionary biology.

## Introduction

Over the past decade, meta-research (or metascience) has emerged as the term for the rigorous evaluation of research [1]. The emergence of meta-research is related to discussions of replication and reproducibility across multiple disciplines, notably psychology [2], and including ecology and evolutionary biology [3–6]. Replication is one focus of meta-research studies in ecology and evolutionary biology [7, 8], but the remit of meta-research encompasses topics such as the extent of selective reporting and publication bias in ecology [9, 10], the prevalence of questionable research practices amongst ecologists [11], and analytic flexibility [12]. Closely related to meta-research studies identifying such problems are works and initiatives proposing solutions, based on principles of openness and transparency. Initiatives in the field include the Tools for Transparency in Ecology and Evolution [13], which was followed by the formation of the Society for Open, Reliable, and Transparent Ecology and Evolutionary Biology (SORTEE) for ecologists and biologists with an interest in transparency and open science [14].

created made use of data and code files shared alongside the published meta-analysis articles surveyed in this study; these data/code files are not included in the above Zenodo repository.

**Funding:** SK received support from a Melbourne Research Scholarship (https://scholarships. unimelb.edu.au/awards/melbourne-research-scholarship) and an Australian Government Research Training Program (RTP) Scholarship (https://www.education.gov.au/research-block-grants/research-training-program). FF received funding from Australian Research Council Future Fellowship FT150100297 (https://www.arc.gov.au/). The funders had no role in study design, data collection and analysis, decision to publish, or preparation of the manuscript.

**Competing interests:** The authors have declared that no competing interests exist.

This study contributes to the meta-research within ecology and evolutionary biology by focusing on computational reproducibility. Computational reproducibility is defined as "obtaining consistent results using the same input data; computational steps, methods, and code; and conditions of analysis" [15, p.46]. By this definition, availability of both the data and code underpinning an article is a necessary prerequisite for computational reproducibility. Thus, our study of computational reproducibility is also a study of data and code availability. (Note that if data but not code were available, recalculation of results could still be attempted using the written description of statistical analysis methods to write fresh analysis code. Such an approach has been called "analytical reproducibility" and has been studied separately [16–18]. Although analytical reproducibility and computational reproducibility are related concepts, in this study we focused on computational reproducibility.)

If we have the shared data and code for an article, then ideally we should be able to use both to recalculate results that match the published results. The technical difficulty of achieving this in practice is well-recognised, even for researchers returning to their own computer code years later [19]. Thus, there have been a number of studies across different disciplines gauging how often results in the published literature can actually be computationally reproduced from data and code. Stodden et al. [20] evaluated the effectiveness of the data and code sharing policy implemented in the journal *Science* in 2011, by attempting to obtain data and code for 204 articles published after the policy change in order to computationally reproduce their results. They obtained data and code for 44% of articles in the sample and were able to successfully reproduce results for 26% of the sample. Wood et al. [21] assessed the computational reproducibility of 109 articles published in 2014 from journals in development, economics, and public health. Their study, described as a "research audit exercise" found that a lack of available data and code meant that reproduction could not be attempted for 71% of articles in the sample. They were able to reproduce results identical to or within rounding of the original results for 27 articles, and found only minor differences in another 5 articles. In psychology, Obels et al. [22] considered a set of 62 Registered Reports published over 2014–18, and found 36 (58%) had shared data and code, making them suitable candidates for attempting computational reproducibility. They successfully reproduced the main results of 21 of these 36 articles, which was 58% of the attempts made. More recently, Crüwell et al. [23] audited 14 articles published in a 2019 issue of *Psychological Science*, all of which had been awarded an Open Data Badge (https://www.cos.io/initiatives/badges) signifying that the article authors had shared the data for reproducing their results. Crüwell et al. [23] found that while all 14 articles did share data, only 6 shared code. Their attempts to computationally reproduce results from this issue found that one article was exactly reproducible, and three were reproducible with only minor differences. In ecology and evolutionary biology, ArchMiller et al. [24] attempted to computationally reproduce a sample of 80 studies published in the *The Journal of Wildlife Management* and *Wildlife Society Bulletin*. They were able to obtain data and code for 19 of the studies, and mostly or fully reproduce the results for 13 of them.

Such results reinforce the centrality of data and code sharing to computational reproducibility. Data sharing is a well-established topic in ecology and evolutionary biology, with numerous efforts to facilitate and improve data sharing, coming from both individual researchers and institutions such as journals. Journals have recognised and stressed the importance of data archiving [25–27]. Researchers have created guides and compiled advice for how to best approach data archiving and sharing [5, 28, 29]. There have also been efforts to review the effectiveness of data archiving policies and assess how the field is doing [30–32]. Code availability in ecology and evolutionary biology has also been studied: Mislan et al. [33] surveyed 96 ecology journals in 2015, and found that only a small minority (14%) required code to be made available alongside published articles (in contrast to 38% of

journals requiring data be made available alongside published articles). Culina et al. [34] repeated this survey in 2020 and found that of the same 96 journals, 75% mandated or encouraged making code available. However, despite this now common journal policy, Culina et al. [34] also found that only 27% of a sample of 346 ecology articles published 2015–19 actually shared code.

### Aims and scope

We conducted computational reproducibility attempts on a sample of meta-analyses published in ecology and evolution journals over 2015–17 (the restriction to meta-analyses is explained in Section 1 of S1 Appendix). Our focus was on using shared data and code files to reproduce specific results. The primary outcome of the reproducibility attempts is the calculation of an overall computational reproducibility "success rate", similar to Stodden et al. [20].

This study commenced in late 2017 following rising interest in meta-research within ecology and evolutionary biology, including interest in data- and code-sharing specifically [32, 33]. While the results of this study are not a reflection of what the rate of computational reproducibility in more recent ecology and evolutionary biology literature might be, they do provide a benchmark of the state of computational reproducibility during the period 2015–17, and provide a point of comparison for other evaluations of computational reproducibility over earlier or more recent periods.

We surveyed the data- and code-sharing rates of the applicable meta-analysis literature. We only counted as "shared" data/code that was reported as already available, rather than data/code that was (potentially) available upon request. It is possible that some authors of the meta-analyses included in this study may have shared their data and code in response to a request. However, a request for data or code requires an interaction between the requesting party and the article authors, and there is a possibility that the request will not be successful, for a variety of reasons (e.g., the authors are no longer contactable via the contact details provided in the article, the authors do not respond in a timely manner, the authors respond but refuse for some reason, or the authors respond but can no longer find the data and code). We decided not to request data or code from article authors in this study, because requesting data/code would introduce a element of the study that may not be reproducible by others: the success or failure of any requests would rely on factors such as timing, existing connections (of lack thereof) with authors, and the purpose behind the request.

### Materials and methods

Our study had four stages: first, we obtained a sample of published meta-analyses from ecology and evolution journals; second, we assessed each meta-analysis for data- and code-sharing; third, we selected results to be reproduced using the shared data and code; and finally, we attempted to reproduce the selected results.

We curated a set of meta-analyses to survey by conducting a Scopus abstract and citation database search (see details in Section 2 of S1 Appendix). The search query, conducted on 20th December 2017, searched article titles, abstracts, and keywords for the string "meta-anal*", subject to two constraints. The first constraint restricted results to articles published between 2015 and 2017, inclusive. The second constraint restricted results to articles published in one of 21 ecology and evolution journal titles (identified by ISSN), which are the same journal titles as used for the survey of meta-analyses conducted in Nakagawa and Santos [35].

The search yielded 229 results. One irrelevant result was found to have been included in the results due to a Scopus database error and was immediately excluded, leaving 228 results.

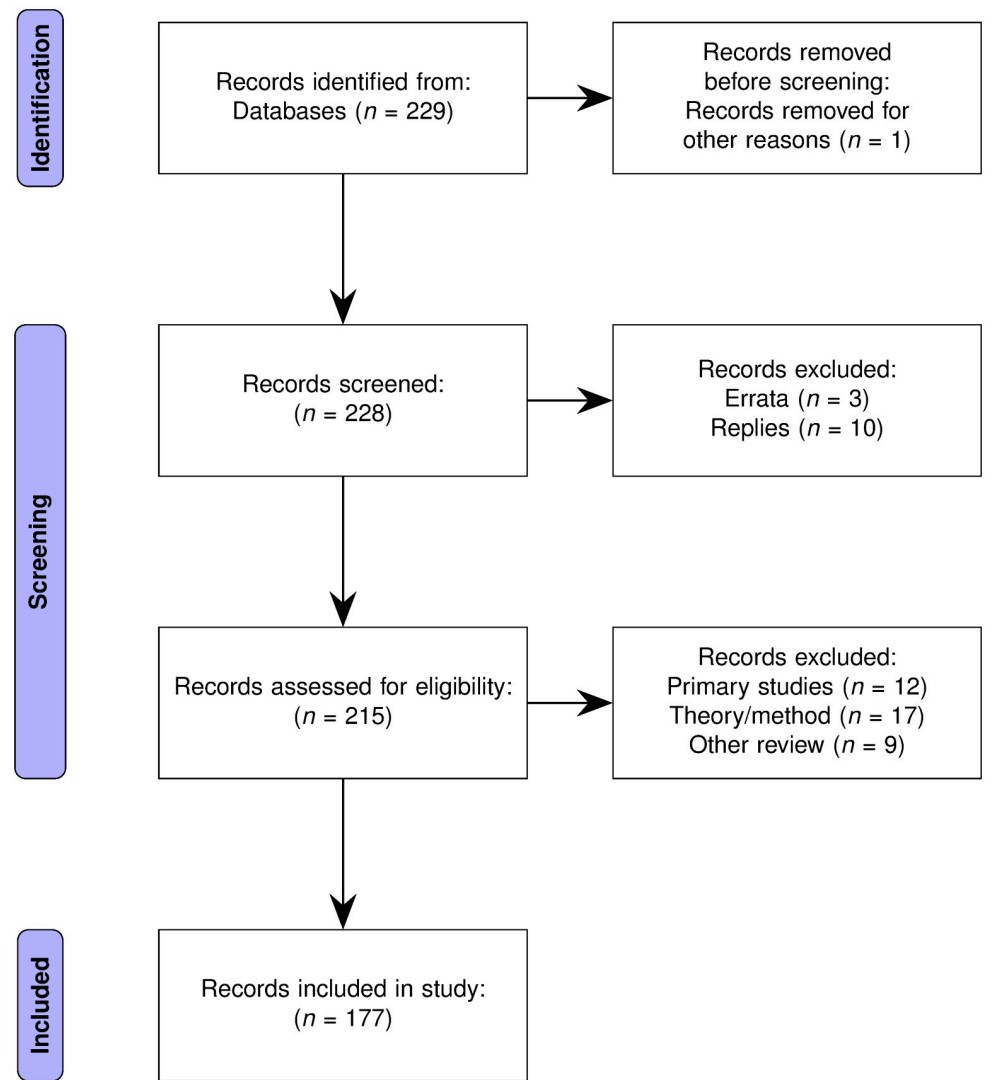

**Fig 1. PRISMA-style flow diagram depicting the article selection process.**

The search results were coded to retain only those articles which were actual meta-analysis studies, details of the coding scheme used are in Section 2 of S1 Appendix. The final set of ecology and evolutionary biology meta-analyses, the basis of the rest of this study, was a set of 177 articles coded as reporting to be meta-analyses. Fig 1 shows a PRISMA-style flow diagram for this study.

### Recording code and data sharing in each article

Each meta-analysis article in the set of 177 was assessed for data and code sharing using the coding scheme detailed in Section 4 of S1 Appendix. It was expected that "data" (curated, formatted information—both numeric and text-based—that was the "raw material" for reported calculations and analyses) would be presented in one or more formatted computer files (e.g., in comma separated values format), possibly accompanied by additional computer documents containing metadata or explanations of the data files' contents. Following Mislan et al. [33]

and Culina et al. [34], we regarded "code" as referring to computer code, specifically analysis code, designed to do tasks such as importing and manipulating data and performing statistical calculations based on data (e.g., calculating summary statistics or fitting models). Code may have been written in a programming language (e.g., R or Python) or it may have been syntax designed to be run by a dedicated statistical analysis software package such as SPSS, SAS, or STATA.

"Sharing" meant that the authors of the article had made data and code files available alongside the publication of the article. For journals which were not Open Access, data and code files provided as supplementary materials on publishers' websites were typically hidden behind subscriber paywalls along with the articles and were not available to everyone. We made the decision that data and code provided in this way counted as having been shared for the purposes of this study. It is for this reason that we have chosen to refer to "shared" data and code rather than "open" data and code, since "open" carries with it connotations about availability and accessibility that may not apply to data and code files provided as supplementary material behind a publisher's paywall.

We also reviewed the methods section of each article for references to the use of software. If an article did not report any details of software used, we reviewed supplementary documentation if supplied. The review process is detailed in Section 5 of S1 Appendix.

## Selecting target results for computational reproduction

For each article in the subset of meta-analysis articles with both shared data and code, we identified a numeric "target" result that would be the basis of the computational reproduction attempt. Selecting a single result from an article involved subjective judgment, and could potentially be manipulated to increase or decrease the chance of success of reproducing each result. To mitigate this risk, we used the following process to identify a target result: our target result would be the first meta-analytic summary effect (consisting of the point estimate, the sample size, and the measure of uncertainty such as a confidence interval) reported in the results section of each article. The reasoning for this strategy is as follows: (i) in general, summary effects are commonly reported in meta-analyses, and so this would identify like results across articles; and (ii) identifying the first reported result is a consistent method of selection across articles that minimises (but does not eliminate) the need for interpretation and therefore reduces the risk of bias. A procedure which allowed for results to be deliberately chosen for computational reproduction could potentially be selected on the basis of perceived ease of reproduction.

In practice, identifying and extracting the first reported meta-analysis summary effect was complicated by two factors. First, articles presented results in different ways: some articles reported results in the body of the text while others referred to a table or figure. We extracted numerical values directly from in-text results and from results presented in tables. For results presented graphically in figures, we extracted numerical results using the software package WebPlotDigitizer version 4.4 for the Windows platform. We rounded all values extracted from figures to two decimal places. Figures required additional interpretation if they plotted multiple summary effects. In these cases, we prioritised extracting the "overall" summary effect if it existed, and otherwise selected the "first" plotted result, according to the layout of the figure (e.g., either the leftmost or topmost result). Frequently, a result was reported in-text and also expressed in a figure/table; we prioritised extracting in-text results over results reported in figures/tables. The second factor was that not all meta-analysis articles actually reported a summary effect result. In these cases, we extracted numerical values for the first-reported result of any kind associated with the meta-analysis.

### Reproducing results and results comparison

For each article, we assessed the shared data and code for its relevance to the identified target result using the following strategy: (i) where available, we consulted documentation accompanying the data and code files; (ii) we examined any comments made within the code syntax files; (iii) where available, we examined the metadata of data files; (iv) we examined the contents of data files directly, looking for clues in variable names and data formats; (v) we examined the syntax of code files directly, looking for clues in function names and the kinds of calculations made. This approach was sufficient to discern with confidence whether the data and code files were applicable to the re-calculation of the target result. We went ahead with attempting to reproduce the target result for each article where both the shared data and code were found to be relevant.

In cases where the code and/or data was not relevant to the identified target result, we stopped attempting to reproduce those particular target results. Rather than do nothing further with these cases, we returned to the article and identified an alternative target result that was relevant to the shared data and code and reported the results of these reproduction attempts separately.

Each reproduction attempt was packaged as a reproducible document written in RMarkdown contained within a controlled computational environment using Docker (details are in Section 9 of S1 Appendix). Where code could be successfully run, reproduced target results were compared with the originally published values. For each target result (which consisted of a set of numbers e.g., summary effect estimate, confidence interval bounds, and sample size), we followed the method used in Hardwicke et al. [17] and quantified the difference between the original published value and reproduced value by calculating the relative error, expressed as a percentage: $\delta = 100 \times |x_R - x_O|/|x_O|$, where $x_O$ is the original reported result value and $x_R$ is the reproduced result value. Note that the relative error is undefined when the original value is zero, and can have a large value when $|x_R - x_O|$ is greater than $|x_O|$. Following Hardwicke et al. [17], we distinguished between three categories of error, exact matches ($\delta = 0\%$), minor numerical discrepancies ($0\% < \delta < 10\%$) and major numerical discrepancies ($\delta \geq 10\%$). Although we calculated the relative error for all target values, for reporting purposes we introduced a category of matches to the rounding precision of the original result: if an original result value was 1.51 (reported to two decimal places), we considered reproduced values of 1.50 and 1.52 ($\pm 0.01$) to be matched to rounding precision.

## Results

The 177 meta-analyses were located within the 21 journals as shown in Table 1. The table also shows the total number of articles from each journal returned by the literature search. Note that neither *Evolutionary Ecology* or *The Quarterly Review of Biology* were found to have published any articles which reported to be meta-analyses over 2015–17 (the literature search did not return any results at all from the journal *Evolutionary Ecology*). The journal found to have the most meta-analyses during 2015–17 was *Biological Reviews*, followed by *Oikos*. The meta-analyses in the sample were fairly evenly spread across the three years searched, as shown in Table 2. Note that six articles have a publication year of 2018; these articles had all been published online during 2017 (and so were picked up in the literature search), but at the time of the literature search had not yet been assigned to a journal issue. These six were subsequently published in journal issues dated in 2018. We kept these six journal articles and regarded them as published in 2017.

**Table 1. Breakdown of the 177 identified meta-analysis articles by journal title.**

| Journal Title | Meta-analysis | | Other | | Total | |
|---|---|---|---|---|---|---|
| | N | % | N | % | N | % |
| Biological Reviews | 24 | 13.6 | 5 | 9.8 | 29 | 12.7 |
| Oikos | 22 | 12.4 | 2 | 3.9 | 24 | 10.5 |
| Ecology Letters | 19 | 10.7 | 1 | 2.0 | 20 | 8.8 |
| New Phytologist | 18 | 10.2 | 5 | 9.8 | 23 | 10.1 |
| Ecology | 13 | 7.3 | 9 | 17.6 | 22 | 9.6 |
| Journal of Applied Ecology | 10 | 5.6 | 2 | 3.9 | 12 | 5.3 |
| Molecular Ecology | 10 | 5.6 | 5 | 9.8 | 15 | 6.6 |
| Oecologia | 10 | 5.6 | 1 | 2.0 | 11 | 4.8 |
| Functional Ecology | 9 | 5.1 | 1 | 2.0 | 10 | 4.4 |
| Journal of Ecology | 7 | 4.0 | 0 | 0.0 | 7 | 3.1 |
| Journal of Animal Ecology | 6 | 3.4 | 3 | 5.9 | 9 | 3.9 |
| Ecological Monographs | 5 | 2.8 | 0 | 0.0 | 5 | 2.2 |
| Behavioral Ecology | 4 | 2.3 | 3 | 5.9 | 7 | 3.1 |
| Evolution | 4 | 2.3 | 0 | 0.0 | 4 | 1.8 |
| Journal of Evolutionary Biology | 4 | 2.3 | 10 | 19.6 | 14 | 6.1 |
| Animal Behaviour | 3 | 1.7 | 2 | 3.9 | 5 | 2.2 |
| Behavioral Ecology and Sociobiology | 3 | 1.7 | 0 | 0.0 | 3 | 1.3 |
| Ecological Applications | 3 | 1.7 | 0 | 0.0 | 3 | 1.3 |
| The American Naturalist | 3 | 1.7 | 1 | 2.0 | 4 | 1.8 |
| The Quarterly Review of Biology | 0 | 0.0 | 1 | 2.0 | 1 | 0.4 |
| Evolutionary Ecology | 0 | 0.0 | 0 | 0.0 | 0 | 0.0 |
| *Total* | *177* | *100.0* | *51* | *100.0* | *228* | *100.0* |

## Rates of data and code sharing

When articles were reviewed for data sharing (as per the coding scheme described in Section 4 of S1 Appendix), a clear majority of 78% or 138 meta-analyses indicated that data had been shared in some manner. Despite the positive indication, in five cases data files could not actually be obtained. This meant that the effective data sharing rate among this sample of meta-analysis articles was 75% (133 out of 177).

The rates of code sharing were much lower in comparison to data sharing: we were able to obtain code files for 16% of meta-analysis articles (28 out of 177). This was one less than the number of articles which had indicated code was available. Of the 28 articles with code, 26 had shared data too, meaning that 15% of articles (26 of 177) in this sample shared both data and code. Section 6 of S1 Appendix breaks down data and code sharing rates by journal.

**Table 2. Breakdown of the 177 identified meta-analysis articles by publication year.** Articles with publication year 2017 includes six articles which were first published online in 2017 before being assigned to a journal issue dated in 2018.

| Publication Year | N | % |
|---|---|---|
| 2015 | 56 | 31.6 |
| 2016 | 61 | 34.5 |
| 2017 | 60 | 33.9 |
| *Total* | *177* | *100.0* |

**Characteristics of shared data and code.** Fig 2 lists the locations of the shared data files for the 133 articles. The majority of articles that shared data did so on the journal publisher's website (58%, $n = 77$): in these cases, the data file(s) had been uploaded as supplementary material to the article. The Dryad Digital Repository [36] was the next most common location to share data (35% or 46 articles), followed by the Figshare (8%, $n = 11$) and Zenodo (1.5%, $n = 2$) repositories. One article was judged to have shared the data for its meta-analyses in tables presented within the published article itself: the article mentioned that the effect sizes and other details for all the individual studies included in the meta-analysis calculations were provided across two tables.

Fig 3 shows the types (formats) of data files shared by the 133 articles. The most common format for data files was a Microsoft Excel spreadsheet (44%, $n = 59$); this included both the binary XLS format and the Open XML XLSX format. The next most common format was the comma separated values (CSV) format (25%, $n = 33$). Data in a variety of plain text formats was shared by 15% of articles ($n = 20$): this included files containing phylogenetic data in NEXUS or Newick tree format. A substantial minority of articles shared tabular data in document formats like Microsoft Word Document formats DOC and DOCX (17%, $n = 22$), Portable Document Format PDF (14%, $n = 19$), Hypertext Markup Language HTML (2%, $n = 3$), and one article shared data in Rich Text Format RTF (1%). Two articles shared data files with a binary format: one article shared a data file in RData format, a binary file used by the R language, and one article shared multiple data files in a proprietary binary format associated with data logging equipment.

Table 3 breaks down the type (i.e., language or compatible software environment) of code shared by the 28 meta-analysis articles which shared code. The majority of articles shared R

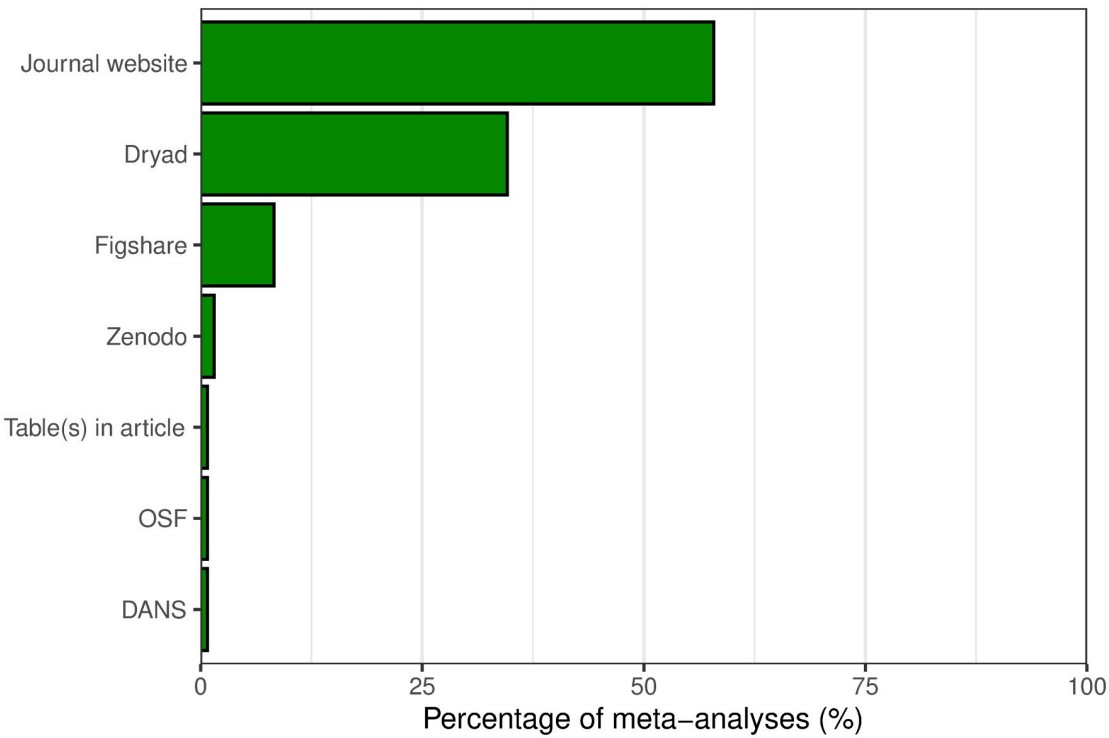

**Fig 2. Breakdown of the locations where articles shared data online.** Note that some articles shared data files in more than one location; both locations were counted, so the percentages indicated add up to more than 100%.

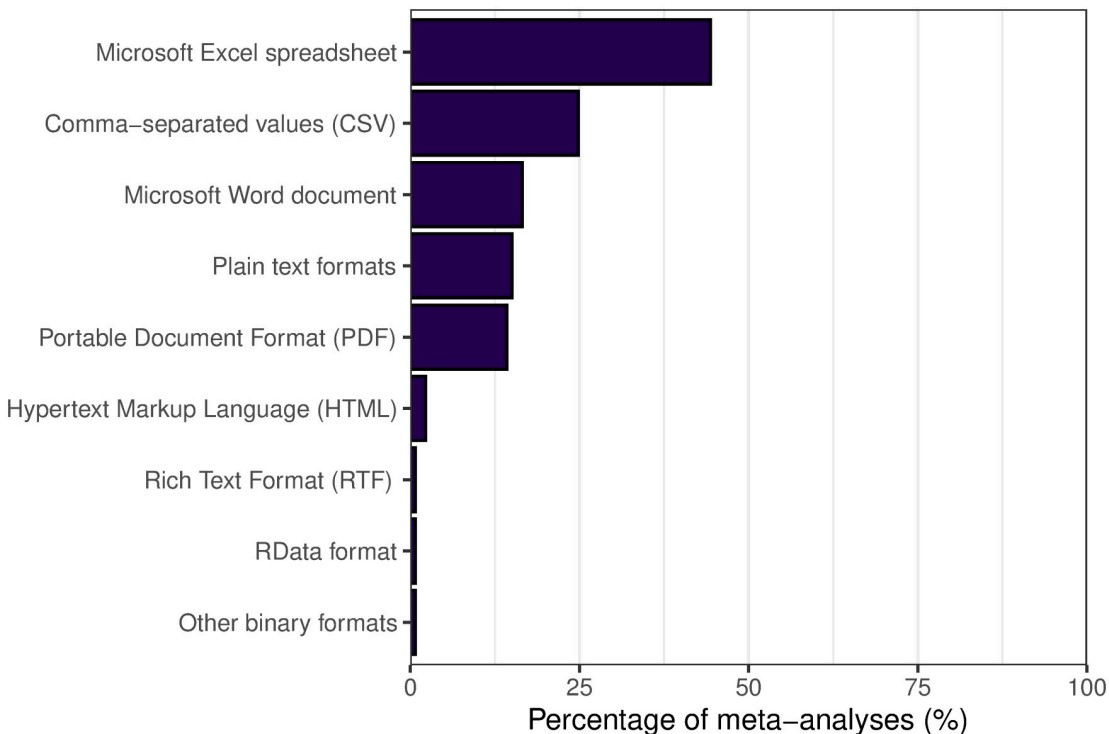

**Fig 3. Breakdown of the types of file format shared by each article.** Some articles shared data files of more than one type, and both types of file were counted (multiple files of the same file format only counted as one). This means that the percentages will add up to more than 100%.

code (26 out of 28, 93%): 25 shared only R code, and one article shared R code and C++ code, which were designed to work together. The remaining two articles shared FORTRAN code and Python code respectively.

## Software mentioned in articles

Overall, 171 meta-analysis articles (97%) mentioned at least one specific software package that was used during the study, whether mentioned in the article text or in supplementary material. The R software environment was the most commonly mentioned software package with nearly 80% of articles mentioning R. The next most commonly mentioned piece of software was MetaWin; 11% of articles mentioned using it. The specialised meta-analysis software package CMA was mentioned by two articles, or 1% of the sample. The full list of all software packages mentioned is in Section 7 of S1 Appendix. Due to the popularity of R in this sample, and the

**Table 3. The 28 code-sharing meta-analysis articles broken down by type of code shared.**

| Type of code shared | N | % |
|---|---|---|
| FORTRAN | 1 | 3.6 |
| Python | 1 | 3.6 |
| R | 25 | 89.3 |
| R and C++ | 1 | 3.6 |
| Total | 28 | 100.0 |

specifics of its package system, R and R packages were summarised separately from the non-R software packages.

There were 144 mentions of software packages that were not the R software environment or an R package. The majority of these mentions were accompanied by a reference: 83 (58%) included a complete citation that appeared in the article's reference section, and 39 (27%) included a short in-text reference. The short in-text references included simple mentions of the software publisher or author, and/or a URL to the software's website. Only 15% of these software package mentions had no citation of any kind. A majority of these software package mentions (95, or 66%) also specified which version of the software package was used.

Nearly 80% (141) of meta-analysis articles mentioned using the R software environment. The majority of these mentions of R included a citation: 86 (61%) included the citation in the reference section and 21 (15%) included a short in-text reference. The version of R used was mentioned in 88 (62%) articles (see Table 9 in Section 7 of S1 Appendix). In total, there were 257 mentions of specific R packages: 220 (86%) included a full citation and 3 (1%) a short in-text reference. The most common R package mentioned was the *metafor* package [37], mentioned by 75 articles (53% of the articles which mentioned R). Package versions were mentioned in 58 (23%) cases. A table listing all R packages mentioned in articles is provided in Table 8 in Section 7 of S1 Appendix.

## Reproducing target results

We used the subset of 26 articles with both shared data and shared code for the reproduction attempts. For each article we selected a target result; in 22 cases, we were able to identify what we termed a "summary effect" result: a mean, correlation, or model parameter such as slope derived from the data collected for the meta-analysis. These target results are detailed in Table 4. In the other 4 cases, the articles did not report such a result, but instead a variety of different results from an eclectic set of analyses. These other results are specified by article in Tables 10–15 in Section 8 of S1 Appendix.

There were 173 separate values across the 26 target results from the articles with both data and code, with an average of 6.7 values making up each target result. This included summary effect estimate values, sample size values, measures of uncertainty such as lower and upper bounds of confidence intervals described in Table 4, and other values described in Tables 10–15 in Section 8 of S1 Appendix.

Table 5 summarises the relevance of the articles' shared code to the target results: Of the 22 articles with summary effect target results, 19 had relevant code and one had partially relevant code. Of the 4 articles with other target results, one had relevant code and two had partially relevant code. The remaining cases did not have relevant code. "Not relevant" meant that the shared code performed calculations or analyses that were unrelated to the calculation of the target result selected for reproduction or any meta-analysis results (the code conducted simulations or analysed experimental data instead.) "Partially relevant" code performed calculations or analyses that related to meta-analysis results, but not the target result selected for reproduction. The "not relevant" and "partially relevant" code could not be used to reproduce the target result.

We judged 20 out of 26 articles with shared data and code (77%) to have code relevant to the target result and attempted to reproduce those 20 results.

We attempted to reproduce the 108 target results associated with the 20 articles with relevant code. The reproduction attempt for each article was fully documented in a report; refer to Section 9 of S1 Appendix for details. We regarded the 65 target results associated with the six articles with irrelevant/partially relevant code as failed attempts (we return to these articles in

**Table 4. Details of the 22 summary effect target results selected for reproduction attempts.** In the table, the following abbreviations are used: CI—confidence interval; HPDI—highest posterior density interval; SE—standard error; n.s.—not stated.

| ID | Study | Result source | Effect size type | N | Estimate | Uncertainty |
|----|-------|---------------|------------------|---|----------|-------------|
| MA016 | [38] | in text (p.1100) | Pearson's $r$ | 49 | -0.83 | <0.001 ($p$-value) |
| MA060 | [39] | in text (p.674) | Fisher $z$-transformation | 37 | 0.044 | (-0.174, 0.289) (95% HPDI) |
| MA062 | [40] | in text (p.1115) | Hedges' $d$ | 37 | -0.205 | (-0.444, 0.035) (95% CI) |
| MA065 | [41] | in text (p.80) | Hedges' $g$ | 703 | -8.42 | (-10.73, -6.63) (95% CI) |
| MA067 | [42] | in text (p.306) | Hedges' $g$ | 52 | -0.21 | 0.07 (SE), -2.7 ($z$-score), 0.006 ($p$-value) |
| MA068 | [43] | in text (p.14) | odds ratio | 75 | 1.82 | (1.37, 2.41) (95% HPDI) |
| MA071 | [44] | Figure 3A (p.538) | response ratio | 50 | -0.26 | (-1.02, 0.51) (95% CI) |
| MA074 | [45] | in text (pp.2795 -2796) | Pearson's $r$ | 43 | 0.183 | (0.089, 0.274) (95% CI) |
| MA081 | [46] | in text (p.5351) | slope parameter | 1296 | 1.30 | (0.95, 1.66) (95% CI) |
| MA091 | [47] | in text (p.2556) | Cohen's $d$ | 65 | 0.56 | (0.42, 0.69) (95% CI) |
| MA095 | [48] | Figure 3A (pp.1495 -1496) | Fisher $z$-transformation | 25 | 0.76 | (0.61, 0.91) (95% CI) |
| MA126 | [49] | in text (p.83) | log odds ratio | n.s. | -1.11 | 0.49 (SE), -2.28 ($z$-score), 0.023 ($p$-value), (-2.06, -0.15) (95% CI) |
| MA145 | [50] | in text (p.366) | Fisher $z$-transformation | 118 | -0.08 | (-0.22, 0.03) (95% HPDI), 38 ($N_{studies}$), 25 ($N_{species}$) |
| MA147 | [51] | in text (p.66–69) | percentage | 49 | 0.13 | 0.030 (SE), (0.074, 0.19) (95% CI) |
| MA155 | [52] | in text (p.565) | Pearson's $r$ | n.s. | 0.51 | 0.01 ($p$-value) |
| MA188 | [53] | in text (p.653) | log response ratio | 818 | -0.363 | (-0.408, -0.318) (95% CI) |
| MA191 | [54] | in text (p.92) | slope parameter | 553 | 0.86 | (0.77, 0.94) (95% CI) |
| MA198 | [55] | in text (p.4595) | Fisher $z$-transformation | 79 | -0.41 | (-0.55, -0.27) (95% CI) |
| MA202 | [56] | in text (pp.1072 -1073) | Hedges' $d$ | 329 | -0.330 | (-0.503, -0.156) (95% CI) |
| MA211 | [57] | Figure 2 (p.374) | log response ratio | 3298 | 0.24 | (0.23, 0.25) (95% CI) |
| MA213 | [58] | in text (p.2004) | difference in means | 654 | -0.07 | 0.362 ($p$-value) |
| MA229 | [59] | Figure 3 (p.256) | log response ratio | 57 | 0.40 | (0.24, 0.53) (95% CI) |

the next section). Table 6 summarises the results of the reproduction attempts of the target results.

Table 6 shows that just under 50% of target results could either be reproduced either exactly or differed only by the rounding precision of the original value (rounding or floating point errors could explain these discrepancies). Of the remaining target results, thirteen differed from the original value by less than 10%, three reproduced values differed from the original value by 10% or more, and there were six target results from three articles that could not be reproduced at all; the circumstances of these six failures are described in Table 7.

The summary of the reproduction attempts in Table 6 counts every target result value separately, whether an effect size point estimate, a lower or upper bound of a confidence interval, or a sample size. Calculating a reproducibility success rate over the total number of values in this way does not consider that the sets of values from each article are inter-dependent, and so the success or failure in reproducing one value from an article may not be considered to be independent of the success or failure in reproducing another value from the same article. The possibility of dependency of reproduction success between the different target values within an article is examined in Section 9 of S1 Appendix.

The original and reproduced values for the summary effect size target results are compared in Table 8. Apart from one failure to reproduce a summary effect size (MA211), the reproduced values were close to the originally reported values. All reproduced summary effect sizes were in the same direction as the original. There were nine exact matches between original and reproduced values. Of those that were not exact matches, six (MA060, MA062, MA071, MA191, MA198, MA229) differed by the rounding precision of the original values, and so

**Table 5. Summary of reviews to gauge the relevance of shared code to each target result.**

| ID | Study | Result type | Code relevance |
|---|---|---|---|
| MA016 | [38] | summary effect | not relevant |
| MA060 | [39] | summary effect | relevant |
| MA062 | [40] | summary effect | relevant |
| MA065 | [41] | summary effect | relevant |
| MA067 | [42] | summary effect | relevant |
| MA068 | [43] | summary effect | partially relevant |
| MA071 | [44] | summary effect | relevant |
| MA074 | [45] | summary effect | relevant |
| MA081 | [46] | summary effect | relevant |
| MA091 | [47] | summary effect | relevant |
| MA092 | [60] | other result | not relevant |
| MA094 | [61] | other result | partially relevant |
| MA095 | [48] | summary effect | relevant |
| MA126 | [49] | summary effect | relevant |
| MA129 | [62] | other result | relevant |
| MA145 | [50] | summary effect | relevant |
| MA147 | [51] | summary effect | relevant |
| MA155 | [52] | summary effect | not relevant |
| MA188 | [53] | summary effect | relevant |
| MA191 | [54] | summary effect | relevant |
| MA198 | [55] | summary effect | relevant |
| MA202 | [56] | summary effect | relevant |
| MA211 | [57] | summary effect | relevant |
| MA212 | [63] | other result | partially relevant |
| MA213 | [58] | summary effect | relevant |
| MA229 | [59] | summary effect | relevant |

were off by ±0.001 (where reported to 3 decimals places) or ±0.01 (where reported to 2 decimal places). Also, five cases with discrepancies (MA060, MA062, MA065, MA198, MA202) used methods which relied on random number generation (Markov chain Monte Carlo and multiple imputation). The code for these articles did not include information about setting a random seed, and so it was not possible to recover the precise target result value as originally calculated by the code.

A full table showing comparisons of original and reproduced values for all target results is provided in Table 16 in Section 9 of S1 Appendix.

**Table 6. Breakdown of the reproduction attempt outcomes for the 173 target results.**

| Outcome of target result reproduction attempt | N | % |
|---|---|---|
| Original and reproduced values match exactly | 75 | 43.4 |
| Original and reproduced values differ by rounding precision | 11 | 6.4 |
| Original and reproduced values differ by less than 10% | 13 | 7.5 |
| Original and reproduced values differ by 10% or more | 3 | 1.7 |
| Failed, could not calculate any value for target result | 6 | 3.5 |
| Failed, code not relevant to target result | 65 | 37.6 |
| Total | 173 | 100.0 |

**Table 7. Descriptions of the failures to reproduce target results.**

| ID | Study | Target result(s) | Description |
|---|---|---|---|
| MA081 | [46] | 2 values (upper and lower confidence interval limits) | The code uses bootstrapping to calculate the reported confidence interval, but we encountered an error: the bootstrapping procedure as coded creates random data from which the bootstrapped value cannot be calculated, making it impossible to complete the bootstrap calculation. |
| MA211 | [57] | 4 values (summary effect estimate, upper and lower confidence interval limits, sample size) | There is a mismatch between the supplied data and code: the code that would clearly calculate the target results attempts to subset the supplied data using a variable that does not appear anywhere in any shared data files. |

## Reproducing target results when code not relevant

The previous section identified six cases where the code shared with the article was only partially relevant or not relevant to the article's meta-analysis results. There were three cases with shared code judged partially relevant, and three cases with shared code judged not relevant (these cases are described in detail in Section 10 of S1 Appendix).

As described earlier, the target results for these articles were regarded as failed reproduction attempts. However, we reviewed the code and data for these articles again, with the following in mind: (i) where the shared code was at least partially relevant to the meta-analysis in the article, could the code that *had* been shared be used to reproduce an alternative meta-analysis target result, and (ii) where the shared code was clearly not relevant to the meta-analysis, was the shared data and meta-analysis methods description in the article enough to allow us to write code to successfully reproduce the selected target result. The results of assessing two

**Table 8. Original and reproduced values of target summary effect sizes, for articles with relevant code.** Percent errors marked with * indicate that these results differed only by the rounding precision of the original values.

| ID | Study | Effect size type | Original | Reproduced | Percent error (%) |
|---|---|---|---|---|---|
| MA060 | [39] | Fisher $z$-transformation | 0.044 | 0.043 | 2.27* |
| MA062 | [40] | Hedges' $d$ | -0.205 | -0.204 | 0.49* |
| MA065 | [41] | Hedges' $g$ | -8.42 | -8.87 | 5.34 |
| MA067 | [42] | Hedges' $g$ | -0.21 | -0.21 | 0.00 |
| MA071 | [44] | response ratio | -0.26 | -0.27 | 3.85* |
| MA074 | [45] | Pearson's $r$ | 0.183 | 0.185 | 1.09 |
| MA081 | [46] | slope parameter | 1.30 | 1.30 | 0.00 |
| MA091 | [47] | Cohen's $d$ | 0.56 | 0.56 | 0.00 |
| MA095 | [48] | Fisher $z$-transformation | 0.76 | 0.76 | 0.00 |
| MA126 | [49] | log odds ratio | -1.11 | -1.11 | 0.00 |
| MA145 | [50] | Fisher $z$-transformation | -0.08 | -0.08 | 0.00 |
| MA147 | [51] | percentage | 0.13 | 0.13 | 0.00 |
| MA188 | [53] | Log response ratio | -0.363 | -0.363 | 0.00 |
| MA191 | [54] | allometric slope parameter | 0.86 | 0.85 | 1.16* |
| MA198 | [55] | Fisher $z$-transformation | -0.41 | -0.42 | 2.44* |
| MA202 | [56] | Hedges' $d$ | -0.330 | -0.340 | 3.03 |
| MA211 | [57] | log response ratio | 0.24 | | |
| MA213 | [58] | difference in means | -0.07 | -0.07 | 0.00 |
| MA229 | [59] | log response ratio | 0.40 | 0.39 | 2.50* |

**Table 9. Breakdown of reproduction attempt outcomes for 59 target results from articles with irrelevant code.**
The irrelevant code shared by four articles (MA016, MA092, MA155, and MA212) required the writing of entirely new code to attempt to reproduce their target results. In this table, "N" refers to the number of reproduction attempts falling into each outcome category, and "%" expresses this as percentage out of all 59 of these attempts.

| Outcome of target result reproduction attempt | N | % |
|---|---|---|
| Original and reproduced values match exactly | 44 | 74.6 |
| Original and reproduced values differ by rounding precision | 1 | 1.7 |
| Original and reproduced values differ by less than 10% | 10 | 16.9 |
| Original and reproduced values differ by 10% or more | 3 | 5.1 |
| Original and reproduced values differ (non-numeric target result) | 1 | 1.7 |
| Total | 59 | 100.0 |

articles fitting scenario (i) are described in Section 10 of S1 Appendix; one article's code despite being partially relevant was judged unworkable and so was treated as part of scenario (ii) along with the three articles with code not relevant.

Table 9 breaks down the outcomes of the analytical reproduction attempts when writing new R code: we were able to calculate a value to compare to the original for all target results from the four articles considered. There were 44 exact matches between original and reproduced values (75%), and of the non-exact matches, one differed by the rounding precision of the original value, ten (17%) reproduced values were within 10% of the original values, and three (5%) reproduced values were more than 10% from the original values. The was also one case of a non-numeric text string not matching the original text string.

As these results show, the reproduction attempts using newly-written R code were largely accurate, even though they did not constitute a computational reproducibility attempt evaluating both the shared data and code of the articles, as was the case for the results in the previous section.

## Computational reproduction success rates

The overall computational reproducibility success rate for this study depends on how it is defined. Different definitions lead to different values for the numerator and denominator in the calculation. We considered the success rate in terms of the number of meta-analysis articles with successful reproductions of the target results. Since multiple target result values were identified in each of the 26 articles with shared data and code, the reproduction success on each individual target result value needed to be collapsed into a single result at the article level. There were different approaches to this, with varying levels of strictness.

Table 10 reports the overall computational reproducibility success rates for different collapsing approaches across two scenarios: (i) when all six code-irrelevant cases were considered failures by default (and thus only the 20 articles with target result-relevant code could be potential successes), and (ii) when the reproduction attempts from both the 20 articles with target result-relevant code *and* the four articles where we wrote new R code were included in the success calculations (the two articles where alternative target results were selected in order to evaluate the shared code were still considered failures by default). In addition, for each scenario, two success rates were calculated: one which expressed the number of successful article reproduction attempts as a percentage of all 177 meta-analysis articles in the sample, and the other which expressed the number of successful article reproduction attempts as a percentage of the subset of 26 meta-analysis articles which shared code and data.

Depending on the level of stringency applied to count as a success, the success rate for the code-relevant cases only was in the range of 4.0–10.7% of all articles in the sample (or 26.9–

**Table 10. Reproducibility success rates at the article level for different collapsing criteria.** In this table, *N* is the number of articles meeting each collapsing criterion, "success rate (%), all" expresses *N* as a percentage of all 177 meta-analysis articles in the sample, and "success rate (%), subset" expresses *N* as a percentage of the subset of 26 articles with shared data and code. In the first three columns of this table, the articles with data and code judged irrelevant to the target results were considered failures by default. In the last three columns, reproduction attempts where we wrote new code to reproduce the target results were included in success calculations.

| Result for article | All code-irrelevant cases considered failures | | | Including attempts where new code was written for code-irrelevant cases | | |
|---|---|---|---|---|---|---|
| | *N* | Success rate (%), all | Success rate (%), subset | *N* | Success rate (%), all | Success rate (%), subset |
| All target result values match original exactly | 7 | 4.0 | 26.9 | 9 | 5.1 | 34.6 |
| At least 50% of target result values match original exactly | 13 | 7.3 | 50.0 | 16 | 9.0 | 61.5 |
| All target result values match original exactly or to rounding precision | 9 | 5.1 | 34.6 | 11 | 6.2 | 42.3 |
| At least 50% of target result values match original exactly or to rounding precision | 17 | 9.6 | 65.4 | 21 | 11.9 | 80.8 |
| All target result values within 10% of original | 15 | 8.5 | 57.7 | 17 | 9.6 | 65.4 |
| At least 50% of target result values within 10% of original | 19 | 10.7 | 73.1 | 23 | 13.0 | 88.5 |

73.1% of articles with code and data). Including the cases where new code was written for the code-irrelevant cases raised the success rate, with a range of 5.1–13.0% of all articles in the sample (or 34.6–88.5% of the articles with code and data).

## Discussion

In their study of the availability of code in ecology, Culina et al. [34] estimated the proportion of the ecology literature surveyed that was *potentially* computationally reproducible. The threshold for articles to be potentially reproducible was that (seemingly) all the code and data required to reproduce results was shared, with the assumption that in practice shared code as well as data was required for reproducibility. They found that 20% of literature published in 2015–16 and 21% published in 2018–19 was potentially reproducible. In this study, we found that 14.7% of articles in our 2015–17 sample (26/177) shared both code and data. Thus, under a definition of computational reproducibility that requires both data and code (used in both Culina et al. [34] and here) we found that 15% of articles had the *potential* to have results computationally reproduced.

Comparing this result to the results in Culina et al. [34] is not entirely like for like, since different sets of journals and time periods were surveyed and this study was restricted to meta-analyses exclusively while Culina et al. [34] was not. Nevertheless, both studies generally agree that the potential for ecology literature to be computationally reproducible was low during the period 2015–17, using the reasonable threshold of 20% as a "low" occurrence rate.

Of course, this study went further than Culina et al. [34] and actually attempted to computationally reproduce results. As seen in Table 10, failures to reproduce results and the discovery that some code was irrelevant resulted in an *actual* computational reproducibility rate of 4.0–10.7% (depending on the criterion for success applied). This actual success rate(s) can be compared with the success rate observed in ArchMiller et al. [24]: 8 out of the 74 suitable articles (published 2016–18) reviewed were found to be fully reproducible, and a further 5 out of 74 articles partially computationally reproducible, for a success rate of 11% (fully reproducible only) or 18% (fully and partially reproducible). (Although 74 out of an original 80 articles were reviewed in total, the researchers could only obtain data and code and thus make a reproducibility attempt for 19 of those articles.) The difference in methods for reporting reproducibility success differed between ArchMiller et al. [24] and this study, making a direct comparison difficult to interpret: ArchMiller et al. [24] rated the computational reproducibility of articles on

a five-point scale which required some qualitative judgment by the researchers, while this study has reported multiple success rates according to different sets of quantitative criteria for success. In addition, in the ArchMiller et al. [24] study, authors of the original articles were contacted to request data and code, which might have contributed towards the higher success rate observed.

In Culina et al. [34], ArchMiller et al. [24], and this study, the low rates of reproducibility (potential or actual) were driven by the low rates of ecology and evolutionary biology articles with both shared data and code. While presenting results in the context of all articles surveyed is clearly warranted, calculating computational reproducibility success rates in this way masks the extent to which data and code, once obtained, can be used to successfully reproduce results. As seen in Table 10, among the subset of articles where computational reproduction was actually attempted, the success rates are much higher as the denominator has been reduced from 177 to 26. Thus, when both data and code were available for an article, *all* target results could be matched *exactly* in 27% of cases. Relaxing the threshold required to rounding precision rather than strictly exact, all target results could be matched in 35% of cases. Although it is still interesting to investigate precisely why the shared data and code do not produce the exact same results more often than this, these results are heartening: the availability of data and code did allow for the exact or close reproduction of results in a substantial fraction of cases. And while this study has included strict criteria for what counts as a success, the level of stringency researchers place on the accuracy and precision of reproduced results will depend on their specific purposes. In a hypothetical circumstance where reproducing all results to within 10% of the original values were acceptable, the clear majority (58%) of articles with data and code in this study would meet this criterion.

The results mentioned above do not include the cases where we wrote new code for those articles where the shared code turned out to be irrelevant to the target result. If these attempts were included in the success rate calculations, the results would improve as shown in Table 10. However, the inclusion of these results as "computational reproducibility" attempts does not fit with our initial definition of computational reproducibility, which posits both data and code be used to recalculate a result. We regarded writing new analysis code from a description of the methods to be a different category of task ("analytic reproducibility"). Conducting analytic reproducibility attempts (based on a sample of the meta-analysis articles which shared data only, for example) in addition to the four attempts in this study was beyond the scope of this study.

Although our canonical computational reproducibility attempts made use of existing code that had been shared to re-run an analysis, we still needed to write bespoke code in order to facilitate the attempt. All attempts required custom code for minor matters like specifying input file locations and re-directing analysis output. Occasionally, custom code was required for more substantial tasks such as processing the shared data files before they could be analysed by the shared code. This frequent need for such additional effort by the researcher conducting the computational reproduction is well recognised in other studies of computational reproducibility. The reproducibility project described in Wood et al. [21] had an expectation that replication code and data received would be "ready-to-run"; they used the term "push button replication" to describe computational reproducibility attempts, which suggests an ideal scenario where an independent researcher can simply "push the button and reproduce the published results" [21, p.2]. However, this was rarely attainable in practice, and to get code working, researchers sometimes had to escalate from minor code troubleshooting (e.g., installing required libraries, or changing the version of a software package used) to "[changing] commands in Stata to allow the code to run, updating commands to the current version of the software, and even correcting typos in an attempt to reproduce the original results" [21, p.7]. This was recognised separately in Stodden et al. [20], who classified the different levels of effort

required when attempting to reproduce results from 22 articles. The classification captured the escalation of effort required from minor difficulties or tweaks (such as installing required software libraries, or adjusting code to work on a different computational system) to major, tedious difficulties (such as needing to write code to re-format data or fill in missing steps) [20]. We encountered similar difficulties to those described in Stodden et al. [20] and Wood et al. [21], and although we have sought to make a clear distinction between computational reproducibility and analytic reproducibility by contrasting "running existing code" with "writing new code", we acknowledge that in practice this distinction may become blurred in cases of computational reproducibility attempts requiring new code to be written. Further scrutiny of the definition of "computational reproducibility" in the light of the results of this study is included in Section 11 of S1 Appendix.

## Limitations

A limitation of this study is that the observed rate(s) of computational reproducibility were possibly underestimated. By design, this study did not attempt to contact article authors seeking access to data and code. Although other studies [20, 24] report mixed success with receiving data and code from authors, it is still the case that assistance from original authors could have lifted the rate of obtained data and code for articles, and in turn potentially the overall reproducibility rate(s).

We did not record the time spent on each reproduction attempt, despite some attempts taking much longer than others. Given that researcher time, effort, and opportunity cost are important considerations, this is perhaps a lost chance to have provided additional information about the activity of reproduction.

Although the strategy of selecting only a single target result to reproduce per article made it feasible to attempt to reproduce results from more articles, it did not provide a measurement of the reproducibility of entire articles. Thus, on the basis of these investigations we cannot claim that any of these articles are entirely "reproducible". Despite this limitation this strategy can be considered in the context of a "triage" approach: a hypothetical article identified as failing such a relatively simple reproducibility check likely has issues with the data, code, or the reliability of published results that must be addressed before further time/effort is expended, or before any results are taken to be accurate for particular purposes.

## Conclusion

This study, like Wood et al. [21] and Crüwell et al. [23], is an example of an audit of the computational reproducibility of the literature that ought to be a regular, ongoing part of the broader project of meta-research to bolster the credibility of results within disciplines. Such checks are an effective gauge the efficacy of data- and code-sharing practices and policies, as well as providing assurance on the accuracy of published results. Our methods for conducting the reproduction attempts can be used as a template for computational reproducibility projects, and which can be expanded upon as required. Our results can be a benchmark and point of comparison for the success rates of other computational reproducibility attempts, at other times and for different types of studies.

We reported the success rate of computational reproducibility of one type of study (meta-analysis) published during 2015–17. The low rate of code sharing among articles published during this period was the principal limitation on the number of possible reproduction attempts. From this, improvement in computational reproducibility would then depend on researchers sharing their code alongside their data when publishing. Journal and funder policies mandating code sharing are clearly one key element of achieving higher rates of code

sharing; another would be to equip researchers with the knowledge they need to produce (re-) usable code that can be shared with confidence. On this point, there are a number of resources specifically for ecology and evolutionary biology researchers. The guide to reproducible code published by the British Ecological Society [64] provides a overview of working reproducibly at all stages of a research project, from initial organisation and structuring of code files to the archiving of a completed project. For the actual writing of code, the introduction to writing "clean code" by Filazzola and Lortie [65] emphasises the formatting and organisation of code to facilitate clear communication of code's purpose and function. There is also an effort to alert researchers to tools that can make reproducible work easier: Braga et al. [66] have compiled a list of 12 ways researchers in ecology and evolutionary biology can use online code repository GitHub, from the straightforward archiving of code and data files to using it to coordinating code development across a team of collaborators.

While widespread availability of code would undoubtedly assist audit studies investigating computational reproducibility post-publication, the success rate of such studies would be further improved (perhaps substantially so) if code was reviewed before publication, perhaps as part of peer review as discussed by Fernández-Juricic [67]. Ivimey-Cook et al. [68] provide a comprehensive primer of code review at all stages of a research project, outlining a workflow for conducting effective reviews. Implementing code review into the research process (whether as part of formal peer review or not) would require a change in current research practices and the allocation of resources; the costs of this would need to be compared against the advantages of enhancing the reproducibility of reported results.

Given the initiatives to improve researchers' code, in concert with journal policies mandating data and code sharing, and the growing awareness of a role for code review, there is reason to be optimistic that future studies of computational reproducibility in ecology and evolutionary biology will not only find higher rates of success, but will be easier for meta-researchers to conduct.

## Supporting information

**S1 Appendix. Additional details about methods and results.**
(PDF)

## Author Contributions

**Conceptualization:** Steven Kambouris.

**Formal analysis:** Steven Kambouris.

**Investigation:** Steven Kambouris.

**Methodology:** Steven Kambouris.

**Supervision:** David P. Wilkinson, Eden T. Smith, Fiona Fidler.

**Writing – original draft:** Steven Kambouris.

**Writing – review & editing:** Steven Kambouris, David P. Wilkinson, Eden T. Smith, Fiona Fidler.

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
