## [Decision Letter · Decision Letter 0]

26 Dec 2023

PONE-D-23-35924Computationally reproducing results from meta-analyses in Ecology and Evolutionary Biology using shared code and dataPLOS ONE

Dear Dr. Kambouris,

Thank you for submitting your manuscript to PLOS ONE. After careful consideration, we feel that it has merit but does not fully meet PLOS ONE’s publication criteria as it currently stands. Therefore, we invite you to submit a revised version of the manuscript that addresses the points raised during the review process.

We look forward to receiving your revised manuscript.

Kind regards,

Elias Kaiser

Academic Editor

PLOS ONE

Journal Requirements:

3. Did you know that depositing data in a repository is associated with up to a 25% citation advantage (https://doi.org/10.1371/journal.pone.0230416)? If you’ve not already done so, consider depositing your raw data in a repository to ensure your work is read, appreciated and cited by the largest possible audience. You’ll also earn an Accessible Data icon on your published paper if you deposit your data in any participating repository (https://plos.org/open-science/open-data/#accessible-data).

SK received support from a Melbourne Research Scholarship (https://scholarships.unimelb.edu.au/awards/melbourne-research-scholarship) and an Australian Government Research Training Program (RTP) Scholarship (https://www.education.gov.au/research-block-grants/research-training-program). FF received funding from Australian Research Council Future Fellowship FT150100297 (https://www.arc.gov.au/). The funders had no role in study design, data collection and analysis, decision to publish, or preparation of the manuscript.

SK received support from a Melbourne Research Scholarship and an Australian Government Research Training Program (RTP) Scholarship. FF received funding from Australian Research Council Future Fellowship FT150100297

SK received support from a Melbourne Research Scholarship (https://scholarships.unimelb.edu.au/awards/melbourne-research-scholarship) and an Australian Government Research Training Program (RTP) Scholarship (https://www.education.gov.au/research-block-grants/research-training-program). FF received funding from Australian Research Council Future Fellowship FT150100297 (https://www.arc.gov.au/). The funders had no role in study design, data collection and analysis, decision to publish, or preparation of the manuscript.

Reviewers' comments:

Reviewer's Responses to Questions

**Comments to the Author**

1. Is the manuscript technically sound, and do the data support the conclusions?

Reviewer #1: Partly

Reviewer #2: Yes

2. Has the statistical analysis been performed appropriately and rigorously? 

Reviewer #1: No

Reviewer #2: Yes

3. Have the authors made all data underlying the findings in their manuscript fully available?

Reviewer #1: Yes

Reviewer #2: Yes

4. Is the manuscript presented in an intelligible fashion and written in standard English?

Reviewer #1: Yes

Reviewer #2: Yes

5. Review Comments to the Author

Reviewer #1: This paper touched upon an interesting topic about reproducibility of ecological and evolutionary meta-analysis papers. The authors found that the code and data sharing in ecological and evolutionary meta-analyses were alarmingly low (26/177; 15%), and only 4-13% of papers were successfully reproduced. Generally, I like the idea of this paper and it is a trendy paper that will arise discussion once formally released. Note that this paper has many merits that I would not mention emphasize them. I will focus on the concerns and the ways that this paper can be improved.

Major comments:

1. The conclusions made are a bit arbitrary and not in a rigorous way to me. They way they calculated the success rate was quite unusual. The core conclusion they made is “The low overall success rate was primarily driven by the low rate of code sharing”. This is a conclusion that any one can make, and does not need any research and analyses. How the author simple equate “no data and code sharing” and “not reproducible ”. A proper way is to use the subset that has data and code available (in this case, 26 papers) to calculate success rate. I assume this paper will reach a quite wide range of audiences and “4-13%” computational reproducibility rate is really conveying misleading information to the community. Literally, it means at best, only 13% meta-analyses can be reproduced in using some software like R; at worst, only 4% can be reproduced. Do the authors believe this?

2. The authors appear to “hacked” the abstract. The authors write in the abstract in a way with the intension to express “how bad of the data sharing and reproducibility is in the field of ecology and evolution”. A decent data sharing rate was found by the authors, as described as “sharing rate among this sample of meta-analysis articles was 75% (133 out of 177).”. This key finding should be emphasized in the abstract. The author said in the abstract: 26 articles (15%) were found to have obtainable data and code files. This sentence gives audience an illusion that only 15% meta-analyses shared data and code. But actually, only 15% meta-analyses shared both data and code at the same time. This is not a transparent way to report your results! Please make it clear.

3. The authors reproduced only meta-analyses with both data and code available, and calculated the success rate. This is not a proper way. What the authors should do is to reproduce the results of meta-analyses with data available, if following the definition of computational reproducibility. Having the original code available is not a criterion to filter your data and use this subset to calculate success rate.

Minor concerns:

Abstract:

The last sentence does not make sense. Please conclude this paper in a broader sense and discuss implications of the results the authors found.

Introduction:

The introduction was not well framed. The authors lacked the descriptions of the recent research progress of meta-science in ecology. The current version only slightly touched upon or simply mentioned meta-science in ecology and evolution. There are quite a few recent work that has not been properly credited. For example

Kimmel, K., Avolio, M. L. & Ferraro, P. J. Empirical evidence of widespread exaggeration bias and selective reporting in ecology. Nature ecology & evolution 7, 1525-1536 (2023).

Yang, Y. et al. Publication bias impacts on effect size, statistical power, and magnitude (Type M) and sign (Type S) errors in ecology and evolutionary biology. BMC biology 21, 1-20 (2023).

The majority of the introduction were about the data and code sharing/archiving. The author’s paper is about reproducibility but not much relevant literature was mentioned, even those in other fields. Without those research progress, it is hard to judge the significance of the paper.

Methods:

The authors used a very outdated dataset. They searched the meta-analysis papers published between 2015 to 2017, on 20th December 2017 in Scopus abstract and citation database search. I would like to suggest the author to update their data. As the open science is a fast-moving moment in ecology and evolution. What the community wants to know is the “current” reproducibility of meta-analyses or the dynamics of the reproducibility. My intuition is that the data and code sharing and reproducibility of recent 5 years’ meta-analyses are expected to increase a lot. The paper is a meta-scientific work. Information/conclusions disseminated in this paper should not have any misleading elements. Imagine that by roughly reading this paper, an audience will have an impression that the success rate of reproducing ecological and evolutionary meta-analyses is 4-13%. This is totally misleading because the way of calculating success rate (a wrong number used as the denominator!) and the outdated data.

I did not get the point of the Box about “Definitions of data, code, and sharing”. Any researchers should know what are the data, code, and sharing, although they might not know the exact definitions. The aim of the Box is to explain the jargons or terminologies that are not familiar by the general audiences.

Although I tried, but I did not find the code and data that can be used to reproduce the paper itself. Please clearly indicate where it is and provide the publicly accessible repository and link.

Results

What is the point of lines 161 – 163 “The practice of including some kind of supplemental information alongside a published article was very common in this sample. The vast majority (168/177, or 95%) of meta-analysis articles included some kind of supplementary or supporting document (regardless of whether or not they also shared data or code).”. Are these sentences relevant to any aims defined in the introduction of this paper?

As said, the author should have a subsection to show the results of reproducing target results when data are available.

Discussions

The discussions were badly written. A proper way is to put your results in the context of the current literature: compare your results with others, interpret your results, the implications, and limitations. The authors only compared their results to one paper and then have a very general discussion on a “topic” that is not very relevant to the topic of their paper – The widespread use of R in ecology and evolution for meta-analysis. Please rewrite the discussions and have proper comparisons.

Reviewer #2: This paper presents the results of a study of the quality of data and code archiving from a sample of meta-analyses in ecology and evolutionary biology.

Understanding the status of data and code archiving in subsets of the literature is important to building an understanding of the implementation of open science practices more generally, and is essential for identifying areas needing improvement.

The methods employed in this study appear appropriate and thorough.

The paper presents sufficient background information, describes methods in detail, provides a thorough accounting of results, and places the work in context.

I have no substantive comments about the science.

The one broad comment I wish to make is that the writing could be improved.

comments below are organized by line number

My first set of comments do not address the quality of the science or the reliability of the presentation (both of which are excellent), but rather the readability of the prose. I make these comments about readability in an attempt to be useful to the primary author, not to demand edits. I think your paper will be better if you take my suggestions, but please invest your time as you see fit.

I provide a few specific suggestions that I hope will serve as examples of edits that can be made throughout the text.

2: Vary your sentence structure. In this case, you start two sentences in a row with “This study”. One possible edit would be to begin the second of these sentences to “We surveyed the data…”

5: Look for superfluous words to cut. In this case, you could say “Twenty-six articles (15%) had obtainable data and code files.”

6: Use active voice. For instance, this sentence could read: “Using these data and code files, we attempted to computationally reproduce the published results.”

Seek to promote coherence between sentences. In other words, help readers link the ideas in sequential sentences, either by using transition phrases (where relevant; examples include ‘Also’, ‘In contrast’, and ‘For example’), or more commonly, by beginning each new sentence with an explicit link to the content in the prior sentence. To see an example of such a transition, see my prior comment where I suggest starting your sentence with “Using these data and code…”, which obviously is a direct link to the data and code you just mentioned in the prior sentence.

161: “-ly” adverbs for emphasis (such as ‘very’) are typically neither necessary nor sufficient to change the reader’s understanding of whatever you’re describing. In this case, saying something was “very common” will not generate a different understanding than saying it was “common”. Instead, the “very” just clutters the sentence.

174-175: avoid including too many moderators before the noun. Too many is typically two or more. In this case, you have “data-sharing” (2) before “articles.” This sentence would read much better as something like “The majority of articles that shared data shared some or all of the data files on the journal publisher’s website.”

Personally, I think it would be even better as something like: “The majority of articles that shared data did so on the journal publisher’s website.”

other miscellaneous comments

51: I encourage you to use the past tense to describe your project. This is a philosophical point more than one of writing style. You planned and implemented the project in the past, and your observations were made in the past, and so the writing should reflect this.

Fig 2 caption: you may wish to avoid contractions

147: It might be helpful to readers define these errors as percentages, and to include the ‘%’ symbol to remind readers of what this number is

174: incomplete sentence

242: the ‘,’ should be replaced by a ‘;’, or else place ‘refer to the Supplementary In243

formation (S9) for details’ in brackets.

Table 9: This table is confusing without consulting the text. The table caption needs a clear explanation of what the ‘N’ and ‘%’ columns refer to.

443 – ‘ans’ typo

6. PLOS authors have the option to publish the peer review history of their article (what does this mean?). If published, this will include your full peer review and any attached files.

Reviewer #1: **Yes: **Yefeng Yang

Reviewer #2: No

---

## [Author Response · Author response to Decision Letter 0]

9 Feb 2024

All responses to reviewers are contained within the Cover Letter (file "Response to Reviewers.pdf"), please refer to this document.

---

## [Decision Letter · Decision Letter 1]

27 Feb 2024

Computationally reproducing results from meta-analyses in ecology and evolutionary biology using shared code and data

PONE-D-23-35924R1

Dear Dr. Kambouris,

We’re pleased to inform you that your manuscript has been judged scientifically suitable for publication and will be formally accepted for publication once it meets all outstanding technical requirements.

Kind regards,

Elias Kaiser

Academic Editor

PLOS ONE

Additional Editor Comments (optional):

Reviewers' comments:

Reviewer's Responses to Questions

**Comments to the Author**

1. If the authors have adequately addressed your comments raised in a previous round of review and you feel that this manuscript is now acceptable for publication, you may indicate that here to bypass the “Comments to the Author” section, enter your conflict of interest statement in the “Confidential to Editor” section, and submit your "Accept" recommendation.

Reviewer #1: All comments have been addressed

Reviewer #2: (No Response)

2. Is the manuscript technically sound, and do the data support the conclusions?

Reviewer #1: Partly

Reviewer #2: Yes

3. Has the statistical analysis been performed appropriately and rigorously? 

Reviewer #1: Yes

Reviewer #2: Yes

4. Have the authors made all data underlying the findings in their manuscript fully available?

Reviewer #1: Yes

Reviewer #2: Yes

5. Is the manuscript presented in an intelligible fashion and written in standard English?

Reviewer #1: Yes

Reviewer #2: Yes

6. Review Comments to the Author

Reviewer #1: The authors did a good job in addressing my comments. Although the analysis is not perfet, it is publicable level.

Reviewer #2: I have read the revised manuscript. The writing is substantially improved. I was confident in the research before, and I remain so.

I have two trivial copy edits:

453 – insert comma after ‘study’

556 – delete first “the”

7. PLOS authors have the option to publish the peer review history of their article (what does this mean?). If published, this will include your full peer review and any attached files.

Reviewer #1: **Yes: **Yefeng Yang

Reviewer #2: No

---

## [Editor Report · Acceptance letter]

4 Mar 2024

PONE-D-23-35924R1 

PLOS ONE

Dear Dr. Kambouris, 

I'm pleased to inform you that your manuscript has been deemed suitable for publication in PLOS ONE. Congratulations! Your manuscript is now being handed over to our production team.

Kind regards, 

on behalf of

Dr. Elias Kaiser 

Academic Editor

PLOS ONE